# Sarcopenia Detection System Using RGB-D Camera and Ultrasound Probe: System Development and Preclinical In-Vitro Test

**DOI:** 10.3390/s20164447

**Published:** 2020-08-09

**Authors:** Yeoun-Jae Kim, Seongjun Kim, Jaesoon Choi

**Affiliations:** 1Biomedical Engineering Research Center, Asan Institute for Life Sciences, Asan Medical Center, Seoul 05505, Korea; lethkim1@gmail.com (Y.-J.K.); tjdwns0318@gmail.com (S.K.); 2Department of Biomedical Engineering, Asan Medical Center, University of Ulsan College of Medicine, Seoul 05505, Korea

**Keywords:** sarcopenia detection, sarcopenia quantification, ultrasound scanning, jacobian, RGB-D camera, force sensor, in-vitro test, ham-gelatine phantom

## Abstract

Sarcopenia is defined as muscle mass and strength loss with aging. As places, such as South Korea, Japan, and Europe have entered an aged society, sarcopenia is attracting global attention with elderly health. However, only few developed devices can quantify sarcopenia diagnosis modalities. Thus, the authors developed a sarcopenia detection system with 4 degrees of freedom to scan the human thigh with ultrasound probe and determine whether he/she has sarcopenia by inspecting the length of muscle thickness in the thigh by ultrasound image. To accurately measure the muscle thickness, the ultrasound probe attached to the sarcopenia detection system, must be moved angularly along the convex surface of the thigh with predefined pressure maintained. Therefore, the authors proposed an angular thigh scanning method for the aforementioned reason. The method first curve-fits the angular surface of the subject’s thigh with piecewise arcs using D information from a fixed RGB-D camera. Then, it incorporates a Jacobian-based ultrasound probe moving method to move the ultrasound probe along the curve-fitted arc and maintains radial interface force between the probe and the surface by force feedback control. The proposed method was validated by in-vitro test with a human thigh mimicked ham-gelatin phantom. The result showed the ham tissue thickness was maintained within approximately 26.01 ± 1.0 mm during 82° scanning with a 2.5 N radial force setting and the radial force between probe and surface of the phantom was maintained within 2.50 ± 0.1 N.

## 1. Introduction

Sarcopenia is a condition characterized by the loss of muscle mass and strength [1]. Since it is strictly related to musculoskeletal mass and strength loss, sarcopenia patient can suffer physical disability, falls, fractures, poor quality of life, and even death. In addition, sarcopenia can cause metabolic problems such as sarcopenic obesity [2]. It is estimated that 5∼13% of elderly people aged 60∼70 years suffer from sarcopenia [3].

To measure muscle mass and strength and to diagnose sarcopenia, various methods have been suggested and clinically applied [4]. These methods include anthropometry, bioelectrical impedance analysis (BIA), dual energy X-ray absorptiometry (DEXA), computer tomography (CT)/magnetic resonance imaging (MRI), and sonography. In anthropometry [5,6,7], an observer directly measures the subject’s body status. It is a simple and clinically applied method in many hospitals. However, it is highly dependent on the observer’s diagnosis variation. BIA [8,9,10] passes a current through the subject’s body and measures the electrical impedance with an electronic circuit. BIA is not sensitive to the observer’s diagnosis variation. However, its accuracy depend on the subject’s hydration status. In DEXA [11,12,13], two X-rays are emitted at the subject’s body and the relative attenuation of two different energy X-rays is measured to determine the subject’s body composition, such as fat, bone mineral, and lean tissue. DEXA is a relatively low-cost method compared to CT or MRI. However, it is not feasible to as a routine test because a patient is required to travel to a center and be treated by a specialist. CT/MRI [14,15,16] is a very accurate modality for diagnosing sarcopenia. However, its shortcomings are the high cost and radiation exposure. Ultrasound [17,18,19,20,21,22,23,24,25,26,27] is feasible for a safe routine test and is cost effective. However, the diagnosis is operator dependent, and there have been limited experiences of ultrasound image analysis for sarcopenia.

Among the abovementioned modalities, qualitative sonography is recently considered a sarcopenia screening paradigm [20,21] because of its usability, cost-effectiveness, and harmlessness to body. Minetto et al. [22] compared muscle thickness and mass of the quadriceps group, obtained by sonography for sarcopenia. Strasser el al. [23] measured muscle thickness, pennation angle in the quadriceps, and rectus femoris for sarcopenia diagnosis by ultrasound imaging and they concluded that it could be an accurate bedside tool for sarcopenia diagnosis and course of sarcopenia in neuromuscular unimpaired patients. Hida et al. [24] compared thigh muscle thickness measurement using sonography with BIA. They used TMT(sum of the rectus femoris and vastus intermedius muscle thickness), which was significantly reduced in subjects with sarcopenia in both gender. Wang [25] performed ultrasound measurements of muscle thickness (MT), fat thickness (FT), MT/body mass index (BMI), and MT/FT in gastrocnemius muscle and they concluded that those with a gastrocnemius MT < 1.5 cm are considered as low muscle mass. Caresio et al. [26] developed an algorithm called “MUSA” for measurement of muscle thickness on longitudinal ultrasound images acquired from different skeletal muscles. It was tested on a database of 200 B-mode ultrasound images of rectus femoris, vastus lateralis, tibialis anterior and medial gastrocnemius and achieved 100 % segmentation success rate.

Ismail et al. [27] used DEXA and diagnostic ultrasound to estimate lean body mass in the trapezius brachioradialis, deltoid pectoralis, and rectus femoris. Berger et al. [28] also showed that radio frequency ultrasound measurements correlated significantly with lean body mass assessed through DEXA.

Watanabe et al. [17] reported a correlation between echo intensity (EI) and muscle strength of rectus femoris and discovered that the value of EI was affected by the thickness of the tissue. However, they reported no clear criterion for EI value because it is highly device dependent.

Moreover, there were many developed devices/methods worldwide to make the ultrasound image more stable and visible by interaction force control, even though they were not intended for sarcopenia diagnosis.

Gilbertson et al. [29] developed a hand-held force-controlled ultrasound probe to make the probe–patient contact force constant and validated the probe by simulation and in-vitro experiments. Harris-Love et al. [30] developed a force feedback image acquisition system with a 6 axis robot [31] to investigate the reliability of the system among examiners with varied experiences. Chatelain et al. [32] presented a confidence-driven control method for an ultrasound probe. They created a confidence map, which represents the image quality difference in the region of interest, controlled the probe orientation using confidence map, and maintained the interaction pressure between the probe and skin with force control. Mathiassen et al. [33] developed a unified 6 axis robot [34] with a force sensor to control the robot scan along the patient’s skin while maintaining an interaction force. They used hybrid velocity-force control to realize the control objective. Victorova et al. [35] realized a scoliosis detection system by utilizing an UR-robot and force sensor.

In addition, there were non-force control-based automatic ultrasound system or methods developed worldwide. Sasaki et al. [36,37] developed a compact portable ultrasound robot for home healthcare. The three degrees of freedom (DOF) robot makes the target image respiratory-free by moving the robot relative to respiratory motion in the ultrasound image. Graumann et al. [38] developed an ultrasound scanning trajectory planning with MRI volume by RGB-D camera point cloud registration and cosine function-based point fitting. They used the k-nearest neighbors(kNN) algorithm [39] to calculate the normal vector of the patient’s surface point cloud and principal component analysis (PCA) [40] to determine the scan direction. Meng et al. [41] proposed a mirror ultrasound scanning method with a RGB-D camera for venous thrombosis detection. In the proposed method, the robotic ultrasound scanner mirrors the sonographer’s scanner position and/or orientation.

In the aforementioned ultrasound sarcopenia diagnosis studies [17,18,19,20,21,22,23,24,25,26,27,28], no sarcopenia-specific automatic device(system) was developed to accurately quantify muscle mass thickness and strength. Moreover, robotic(automatic) ultrasound systems [29,30,31,32,33,34,35,36,37,38,39,40,41] have not been developed for sarcopenia diagnosis. Researches on RGB-D camera to make use of the point cloud information [38,41] requires intra/pre-operational coordinate registration, image segmentation and processing, which are time-consuming and clinically unfeasible. Therefore, in this study, the authors developed an automatic sarcopenia detection system with a commercial ultrasound probe, RGB-D camera, and force sensor. Moreover, an angular thigh scanning method was proposed to accurately measure the muscle thickness in the human thigh. The proposed method uses piecewise arc fitting of the subject’s angular thigh surface with an RGB-D camera, and no prior subject-specific registration or image analysis is needed. The main contributions of the manuscript are listed as follows.
A four DOF sarcopenia detection system with conventional ultrasound probe, RGB-D camera and force sensor is developed to accurately measure the muscle thickness in the subject’s thigh (i.e., rectus femoris, medial gastrocnemius).An angular thigh scanning method with an ultrasound probe is proposed to angularly scan the surface of the subject’s thigh. First, the method curve-fits the angular surface of the subject’s thigh with an RGB-D camera with piecewise arcs. Second, the ultrasound probe is moved along the fitted arc using the Jacobian matrix. The radial interface force can be maintained by a force feedback control.An in-vitro test with ham-gelatin phantom is performed to validate the system and the proposed method.

The Materials and Methods section explains the developed sarcopenia detection system and the proposed angular thigh scanning method. The Results section presents the ham-gelatin phantom test. The Conclusions and Discussion section discusses the research summary and limitations of the present system and method with future research directions.

## 2. Materials and Methods

### 2.1. Sarcopenia Detection System

#### 2.1.1. Sarcopenia Detection System Overview

The overall schematic of the developed sarcopenia detection system is shown in Figure 1a–c. The figures specifically show the perspective view, front view, and side view of the system, respectively. The developed system consists of an arc-shaped lower part and a commercial ultrasound probe holding upper part. The upper part has a 40 mm radial stroke with a ±30° roll angle (ψ) range, whereas the lower part has a ±70° angular (θ) range with a 190 mm linear stroke, as depicted in Figure 1a. The robot (sarcopenia detection system) coordinates is represented by red colored arrows in Figure 1a–c. The height of the developed sarcopenia detection system is 350 mm with a 200 mm diameter of the lower part. Its central point O is located at the center position of the arc in the lower part and it has X-Y-Z coordinate directions. The black box above the sarcopenia detection system is the RGB-D camera [42] with its camera coordinates X${}_{c}$-Y${}_{c}$-Z${}_{c}$, which is used for image acquisition.

The design objective of the system is to scan the human thigh with the ultrasound probe oriented to the normal vector of the surface of the thigh and ensure the contact force between ultrasound probe and thigh be constant. To fulfill these design requirements, the developed system has 4 DOFs, as depicted in Figure 1a. The θ directional angular movement (±70°) and Z-directional linear movement (190 mm) render the ultrasound probe move along the surface of the thigh in both radial and longitudinal directions. During these movements, the ±30° roll angle (ψ) movement orients the probe to the normal vector of the surface and a 40 mm radial stroke maintains the predefined interface force.

In Figure 2a,b, the actual usages of the developed system are represented. The figures also show that the developed system can be attached to manual or automatic arm, and the subject can lay down (a) or sit down (b) so that the clinician can diagnose sarcopenia using the developed system. If the human thigh is placed below the arch-shaped lower part of the sarcopenia detection system, the scanning process begins by moving the ultrasound scanner.

#### 2.1.2. Details of the Sarcopenia Detection System and Test Section

The test section with the developed sarcopenia detection system is shown in Figure 3a,b. The test section shown in Figure 3a consists of a sarcopenia detection system, image processing PC (Intel(R) Core(TM) i7-3930K CPU, 16.0GB RAM, Windows 10 Enterprise), main control PC (Intel(R) Core(TM) i7-3930K CPU, 16.0GB RAM, Windows 10 Enterprise), RGB-D camera (Orbbec Astra Mini-S [42]), and a motor control box. In Figure 3a, the sarcopenia detection system is placed on the corrugated box and the RGB-D camera is positioned and oriented to view downward. The image processing PC processes the RGB-D camera image frame, and the main control PC controls the sarcopenia detection system. In the motor control box, four motor controllers (EPOS 4, Maxon motors Inc. [43]) are used to position-control the four motors in the sarcopenia detection system.

In Figure 3a, the RGB-D camera is used to acquire the point cloud of the subject’s thigh. The camera has range of 0.35∼1 m, field of view (FOV) of 60°H × 49.5°V × 73°D, RGB/depth image resolution of 640 × 480 @30fps, and accuracy of ±1∼3 mm @1 m. Note that the position and orientation of the RGB-D camera are adjusted to match the orientation and center position of the camera coordinates to that of the robot(sarcopenia detection system) coordinates. The center position of the camera is located at (0,456,0) in robot coordinates to maximize the number of points on the subject’s thigh.

The upper and lower parts of the sarcopenia detection system are shown in Figure 3b. The upper part has two Maxon motors (DCX 22, Maxon Motors Inc. [43], Sachseln, Switzerland) to move the ultrasound probe in a 40 mm radial stroke and a ±30° roll angle (ψ) range. The lower part also has two Maxon motors (DCX 26, DCX 32, Maxon Motors Inc. [43], Sachseln, Switzerland) to move the upper part with a ±70° angular (θ) range with 190 mm linear stroke, as depicted in Figure 1a,b. The radius of the curvature of the angular (θ) movement is 77.5 mm, and the roll directional movement (ψ) rotates with respect to the blue-coloreQd circle, illustrated in Figure 4.

As shown in Figure 1c and Figure 3b, the sarcopenia detection system has a nested structure, in which the roll angle (ψ) realizing mechanism is fixed on the radial stroke mechanism. The radial stroke mechanism is fixed on the angular (θ) directional mechanism, and the angular (θ) directional mechanism is fixed on the Z directional linear mechanism. With the nested structure, each DOF can move independently, and the probe can perfectly scan the thigh. Note that the angular (θ) and roll (ψ) movements are realized by cable-driven mechanical system, as depicted in Figure 4. The position and angular control resolutions of the sarcopenia detection system are 0.1 mm, 0.1°, 0.1 mm, and 0.9° in Z, θ, r, and ψ direction, respectively. Note that the ψ direction angle is controlled by feedback from the AHRS sensor in Figure 4, while others are open-loop controlled.

The sensors installed in the sarcopenia detection system are shown in Figure 4. These are AHRS sensor (myAHRS+, Withrobot, Inc. [44], Seoul, Korea), ultrasound probe (SP-L01, Medical, Interson, Inc. [45], Pleasanton, CA, USA), and F/T sensor (Nano 17, ATI Industrial Automation, Inc. [46], Apex, NC, USA).

As depicted in Figure 4, the AHRS sensor is fixed on the arc-shaped frame to measure the roll (ψ) angle. It is calibrated with software from vendor, and its angular resolution is ±0.005493° with ±180° measurement range. The roll angle controllability error of the system can be suppressed by feedback control with the Attitude and Heading Reference Systems (AHRS). The ultrasound probe scans the subject’s thigh and produce ultrasound images. Its depth range is 1∼100 mm and bandwidth is 5∼10 MHz with 2D, linear array form factor. The ultrasound probe can be directly plugged in the image processing PC with USB 2.0 interface. As shown in Figure 4, the ultrasound probe is held between two polycarbonate plates, and one of the polycarbonate plates is fixed on the F/T sensor. The F/T sensor was used to measure the radial directional force between the probe and thigh during scanning. Its resolution is ±0.003125 N with a ±12 N measurement range. In Figure 4, one side of the F/T sensor is fixed on a polycarbonate plate, which is attached to the roll angle movement aluminum plate.

#### 2.1.3. Communications of the Sarcopenia Detection System

The communication wiring of the sarcopenia detection system is shown in Figure 5. In the upper-right corner of the figure, four communication protocols that are used in the system are listed. The RGB-D camera, AHRS sensor, and ultrasound probe are connected to the PC via USB 2.0. The F/T sensor output is connected to the F/T sensor converter via device-specific protocol, and the F/T sensor converter is connected to the main control PC via ethernet. The four motors in the sarcopenia detection system are connected to four motor controllers in the motor control box, and the motor controllers are connected to the main control PC by the CAN interface. The image processing PC is connected to the main control PC via ethernet.

### 2.2. Angular Thigh Scanning Method

The developed sarcopenia detection system must scan the human thigh with ultrasound probe oriented normal to the surface of the thigh and make constant contact force during θ directional scanning. The muscle thickness in the subject’s thigh is assumed to be constant during θ directional thigh scanning when the aforementioned requirements are fulfilled. Therefore, the angular thigh scanning method, which fufill the requirements, was proposed. The proposed method can be divided into two sub-methods. One is planar piecewise arc curve fitting of the convex thigh surface, and the other is Jacobian-based ultrasound probe moving. Each method is explained in the following subsections. Note that complete thigh scanning must include Z-directional movement with θ directional scanning. However, in this research, only the θ directional scanning method is proposed at the specified Z position.

#### 2.2.1. Piecewise Arc Curve Fitting

The subject’s upper thigh is convex when placed in the sarcopenia detection system, as depicted in Figure 2. The convex curve of the upper thigh at a specified Z position, which lies in the X-Y plane in Figure 1, must be determined for the θ directional ultrasound scanning. However, the depth data from the RGB-D camera [42] are pointwise and sparse. Hence, smooth surface fitting is required to move the ultrasound probe continuously. Therefore, a planar piecewise arc curve fitting of convex depth points of the subject’s upper thigh is proposed. The proposed method models the convex curve of the subject’s upper thigh with one or several piecewise arcs, which is sufficient for describing the continuous arc of the subject’s upper thigh. The depth points for curve fitting are extracted from the RGB-D camera. A point cloud example obtained from the RGB-D camera [42] in Figure 3a is depicted in Figure 6a,b. The figures represent the image and depth data when the right thigh of the subject is captured from the camera. Note that in depth data in Figure 6b have black regions, which is not feasible for data processing.

The preliminary test points for piecewise arc curve fitting were extracted from the image stream of a pig doll, as depicted in Figure 7. A sky blue-colored pig doll, which has approximate diameter of 130 mm and length of 500 mm, is inserted in the sarcopenia detection system, and RGB-D data were collected using a data acquisition program provided by the camera vendor [42]. The cut plane, which is represented as a red line in Figure 7b, indicates the X-Y plane to extract planar convex arc points. The cut plane is located at (0,0,100). Note that the point data in camera coordinates(X${}_{c}$-Y${}_{c}$-Z${}_{c}$ in Figure 1) were converted to robot coordinates(X-Y-Z in Figure 1) by a transformation matrix between the camera and robot coordinates for data integrity with the sarcopenia detection system.

Figure 8a,b represent the collected depth data in the preliminary test in Figure 7. The collected data are displayed with a three-dimensional (3D) data display program, and Figure 8a and Figure 8b represent X-Z plane and X-Y plane views of the 3D displayed data, respectively. The cut plane in Figure 7b is also represented in Figure 8a, while the red circle in Figure 8b represents the convex back of the pig doll in Figure 7. A total of 32 points were extracted to represent the convex surface points of the back of the pig doll in the Z = 100 mm. The outer regions of the extracted points in ±X direction are not feasible and are depicted as a black region in Figure 7b. Note that the inter-point distances are not uniform in Figure 8c.

The proposed piecewise arc curve fitting method fits the extracted data points, as depicted in Figure 8c, to several circular arcs. Algorithm A1 in Appendix A represents a pseudo-C code of piecewise arc curve fitting. Algorithms A2 and A3 represent the sub-functions used in Algorithm A1. In Algorithm A1, the arcs_number, number_of_points_in_each_arc, and surface_points are pre-determined in Data, and the results are the arc_center_point and arc_radius of each arc. Note that surface_points are represented by 3D arrays and the clustering of surface_point into each arc is pre-processed by manual clustering. The blue-colored comments in Algorithms A1–A3 explain the overall behavior of each line.

In Algorithm A1, the for loop in line 1 represents the iteration by the number of arcs. In the loop, initial_radius, initial_angle, and initial_point to start fitting are determined in line 2∼line 4 by two functions, which are represented in Algorithms A2 and A3 in Appendix A. After initializing, the main fitting begins. It changes the initial_position and initial_radius of the arc by ±5 mm with 1 mm increments (line 13∼15) and inserts the changed values to mpfit 1-3(a) algorithm [47], which provides a robust non-linear least squares curve fitting (line 17). The fitting_error_func function in line 9 and 18 calculates the least squares curve fitting error. The optimal arc is determined by comparing the results of the mpfit in three for-loops. After finding each optimal_arc, the outermost loop finishes.

Algorithm A1 is implemented in the image processing PC with Visual Studio 2018 C/C++ compiler. The piecewise arc curve fitting results with the extracted points in Figure 8c are presented in Table 1 and Figure 9. The curve fittings of 1 arc and 2 arc with Algorithm A1 were performed for piecewise curve fitting of the extracted points in Figure 8c. In Table 1, the total error and average error are calculated by Equations (Equation 1) and (Equation 2). In Equations (Equation 1) and (Equation 2), p_ref_ij is the reference points represented in Figure 8c and surface_points in Algorithm A1. p_fitting_ij is the resultant fitting points. Note that the notation i is the arcs_number and j is the point number for each arc, represented in Algorithm A1. Further, M is the number_of_arcs and N is the number_of_points_in_each_arc in Algorithm A1.

In 2 arc curve fitting, first half of the total points (left side points in Figure 8c) were assigned as the reference points in 1st arc and the remaining points were assigned as 2nd arc.
(1)$$total\_error=\sqrt{\sum_{i=1}^{M}\sum_{j=1}^{N}{\left({∥p_{ref\_ij}-p_{fitting\_ij}∥}_{2}\right)}^{2}},$$
(2)$$average\_error=\sqrt{\frac{\sum_{i=1}^{M}\sum_{j=1}^{N}{\left({∥p_{ref\_ij}-p_{fitting\_ij}∥}_{2}\right)}^{2}}{MN}}.$$

The average errors in Table 1 are 0.19 mm in 1 arc curve fitting and 0.04 mm in 2 arc curve fitting indicate that the proposed method is applicable to planar convex curve modeling of the upper thigh and that errors can be compensated by some control algorithm. The graphs in Figure 9a,b compare the reference points and fitting results for 1 arc and 2 arc curve fittings, respectively. In Figure 9a, a small discrepancy between the reference points (violet points) and the 1 arc fitting points (blue points) appeared. However, in Figure 9b a few differences were observed in the arc fitting results between reference points and 2 arc fitting result.

#### 2.2.2. Ultrasound Probe Moving Method

With the piecewise arc curve fitting result, the proposed Jacobian-based ultrasound probe moving method must continuously move the ultrasound probe while the probe oriented to the normal vector of the arc and constant probe contact force during θ directional thigh scanning. The planar kinematics of the sarcopenia detection system, the Jacobian, and the ultrasound probe moving method by the induced Jacobian are sequentially explained.

##### Planar Kinematics of the Sarcopenia Detection System

Figure 10a shows some geometric variables for the planar kinematics of the sarcopenia detection system. The X-Y coordinates in Figure 10a correspond to the X-Y coordinates in Figure 1. The blue arc in Figure 10 represents the fitted arc of the subject’s upper thigh and the gray-colored quadrant and dark gray-colored object in Figure 10a represents the semicircular lower part of the sarcopenia detection system and ultrasound probe. In Figure 10a, the ultrasound probe contacts the fitted arc at P(x,y) of the red-colored dot. The $O^{{}^{\prime }}$ point in the blue-colored dot corresponds to blue-colored circle in Figure 4, which is the center of the ψ directional rotation. The r, θ, and ψ in Figure 10a are the local variables of the sarcopenia detection system whereas x, y, and ϕ are the global position variables of the center of the contact area of the ultrasound probe.

The planar kinematics of the sarcopenia detection system are represented by Equation (Equation 3). Using this equation, the global position(x, y, and ϕ) can be determined by local variables (r, θ, and ψ). Note that H is the distance between $O^{{}^{\prime }}$ and P in Figure 10a. The Jacobian of the right side of Equation (Equation 3) can be calculated by the partial derivation of the right side of this equation with respect to the local variables. It is represented in Equation (Equation 4) and the inverse Jacobian is in Equation (Equation 5).
(3)$$\left[\begin{array}{c} x\\ y\\ \phi \end{array}\right]=\left[\begin{array}{c} -H\cos(\theta +\psi )+(r+H)\cos\theta \\ -H\sin(\theta +\psi )+(r+H)\sin\theta \\ \frac{\pi }{2}-\left(\theta +\psi \right) \end{array}\right]$$
(4)$$\begin{array}{c} \left[\begin{array}{c} \delta x\\ \delta y\\ \delta \phi \end{array}\right]=J\left[\begin{array}{c} \delta r\\ \delta \theta \\ \delta \psi \end{array}\right]=\left[\begin{array}{ccc} \cos\theta & -(r+H)\sin\theta +H\sin(\psi +\theta ) & H\sin(\psi +\theta )\\ \sin\theta & \left(r+H)\cos\theta -H\cos(\psi +\theta \right) & -H\cos(\psi +\theta )\\ 0 & -1 & -1 \end{array}\right]\left[\begin{array}{c} \delta r\\ \delta \theta \\ \delta \psi \end{array}\right] \end{array}$$
(5)$$\begin{array}{c} \begin{array}{cc} \left[\begin{array}{c} \delta r\\ \delta \theta \\ \delta \psi \end{array}\right] & =J^{-1}\left[\begin{array}{c} \delta x\\ \delta y\\ \delta \phi \end{array}\right]\\ \; & =-\frac{1}{\left(r+H\right)}\left[\begin{array}{ccc} -(r+H)\cos\theta & -(r+H)\sin\theta & -(r+H)H\sin\psi \\ \sin\theta & -\cos\theta & H\cos\psi \\ -\sin\theta & \cos\theta & \left(r+H)-H\sin(\psi +2\theta \right) \end{array}\right]\left[\begin{array}{c} \delta x\\ \delta y\\ \delta \phi \end{array}\right] \end{array} \end{array}$$

##### Jacobian Based Ultrasound Probe Moving

Figure 10b shows geometric variables for Jacobian-based ultrasound moving method. C(α, β) in Figure 10b is the center position of the blue-colored fitted arc. $P_{0}$∼$P_{N}$ points on the arc were constructed by dividing the fitted arc by δθ. R is the radius of the fitted arc. Note that C(α, β) and R are determined by Algorithm A1. The present scanning point is represented as a green-colored dot ($P_{i}$($r_{i}$, $\theta_{i}$, $\psi_{i}$)) and the next scanning point is reprented as a red-colored dot ($P_{i+1}$($r_{i+1}$, $\theta_{i+1}$, $\psi_{i+1}$)). $P_{i+1}$−$P_{i}$ is δ$r_{i}$, δθ, δ$\psi_{i}$. Note that δ$\theta_{i}$ is a constant, which is written as δθ, omitting i notation. $\psi_{i+1}$ is the rotating angle of the ultrasound probe at the next position $P_{i+1}$ with respect to $P_{i+1}$ point, which is the acute angle between line OW and line $CW^{{}^{\prime }}$. Note that ψ is the rotating angle with respect to $O^{{}^{\prime }}$ in Figure 10a, which is the same as $\psi_{i+1}$ in Figure 10b.

When piecewise arc curve fitting is finished and the initial posture at $P_{0}$($r_{0}$, $\theta_{0}$, $\psi_{0}$) in Figure 10b is calculated using simple geometric relation and kinematics, the center point of the contact area of the ultrasound probe is moved on $P_{0}$($r_{0}$, $\theta_{0}$, $\psi_{0}$) posture. Then, the sarcopenia detection system sequentially moves the ultrasound probe to $P_{1}$, $P_{2}$, …, $P_{N}$ postures by Jacobian-based incremental movements. If the first arc scanning is finished at $P_{N}$, the second arc scanning starts with $P_{0}$ of the second arc, which is the $P_{N}$ of the first arc. To move the ultrasound probe from $P_{i}$ posture to $P_{i+1}$ posture, δ$r_{i}$, δθ, δ$\psi_{i}$ must be determined by previous posture $P_{i}$($r_{i}$, $\theta_{i}$, $\psi_{i}$). It can be calculated by using the following equations.
(6)$$y_{i+1}=\tan\left(\theta_{i}+\delta \theta \right)x_{i+1}$$
(7)$${\left(x_{i+1}-\alpha \right)}^{2}+{\left(y_{i+1}-\beta \right)}^{2}=R^{2}$$
(8)$${\left(x_{i+1}-\alpha \right)}^{2}+{\left(\tan\left(\theta_{i}+\delta \theta \right)x_{i+1}-\beta \right)}^{2}=R^{2}$$
(9)$${x_{i+1}}^{2}-2\alpha x_{i+1}+\alpha^{2}+\tan^{2}\left(\theta_{i}+\delta \theta \right){x_{i+1}}^{2}-2\beta \tan\left(\theta_{i}+\delta \theta \right)x_{i+1}+\beta^{2}=R^{2}$$
(10)$$x_{i+1}=\frac{\alpha +\beta \tan\left(\theta_{i}+\delta \theta \right)\pm \sqrt{{\left(\alpha +\beta \tan\left(\theta_{i}+\delta \theta \right)\right)}^{2}-\left(1+\tan^{2}\left(\theta_{i}+\delta \theta \right)\right)\left(\alpha^{2}+\beta^{2}-R^{2}\right)}}{1+\tan^{2}\left(\theta_{i}+\delta \theta \right)}$$
(11)$$y_{i+1}=\tan\left(\theta_{i}+\delta \theta \right)\frac{\alpha +\beta \tan\left(\theta_{i}+\delta \theta \right)\pm \sqrt{{\left(\alpha +\beta \tan\left(\theta_{i}+\delta \theta \right)\right)}^{2}-\left(1+\tan^{2}\left(\theta_{i}+\delta \theta \right)\right)\left(\alpha^{2}+\beta^{2}-R^{2}\right)}}{1+\tan^{2}\left(\theta_{i}+\delta \theta \right)}.$$

Equations (Equation 6)–(Equation 11) determine $P_{i+1}$($x_{i+1}$, $y_{i+1}$) point in Figure 10b. Equation (Equation 6) is the line equation of the $P_{i+1}$ point and Equation (Equation 7) is a circle equation centered at the α, β point. Note that in Figure 10b, $P_{i+1}$($x_{i+1}$, $y_{i+1}$) is the intersection point of line O$P_{i+1}$ and circle C. If Equation (Equation 6) is inserted in Equation (Equation 7) and solving quadratic equation, $x_{i+1}$ can be determined using Equation (Equation 10). $y_{i+1}$ in Equation (Equation 11) can also be determined by inserting Equation (Equation 10) into Equation (Equation 6). However, only one feasible solution must be selected from the two solutions in Equation (Equation 10) by inspecting the solutions.
(12)$$\vec{\alpha }=\frac{\vec{OW}}{∥\vec{OW}∥}=\frac{1}{\sqrt{{x_{i+1}}^{2}+{y_{i+1}}^{2}}}\left[\begin{array}{c} x_{i+1}\\ y_{i+1} \end{array}\right]$$
(13)$$\vec{\beta }=\frac{\vec{CW^{{}^{\prime }}}}{∥\vec{CW^{{}^{\prime }}}∥}=\frac{1}{\sqrt{{\left(x_{i+1}-\alpha \right)}^{2}+{\left(y_{i+1}-\beta \right)}^{2}}}\left[\begin{array}{c} x_{i+1}-\alpha \\ y_{i+1}-\beta \end{array}\right]$$
(14)$$\psi_{i+1}=\cos^{-1}\left(\vec{\alpha }\cdot \vec{\beta }\right)=\cos^{-1}\left(\frac{1}{\sqrt{{\left(x_{i+1}-\alpha \right)}^{2}+{\left(y_{i+1}-\beta \right)}^{2}}}\left[\begin{array}{c} x_{i+1}-\alpha \\ y_{i+1}-\beta \end{array}\right]\cdot \frac{1}{\sqrt{{x_{i+1}}^{2}+{y_{i+1}}^{2}}}\left[\begin{array}{c} x_{i+1}\\ y_{i+1} \end{array}\right]\right).$$

Equations (Equation 12) and (Equation 13) are unit vectors of vector OW and vector $CW^{{}^{\prime }}$, respectively, to calculate the intersection angle $\psi_{i+1}$. $\psi_{i+1}$ is calculated using Equations (Equation 12)–(Equation 14).
(15)$$\begin{array}{c} \begin{array}{cc} \phi_{i+1} & =\frac{\pi }{2}-\left(\theta_{i+1}+\psi_{i+1}\right)=\frac{\pi }{2}-\left(\theta_{i}+\delta \theta +\cos^{-1}\left(\vec{\alpha }\cdot \vec{\beta }\right)\right)\\ & =\frac{\pi }{2}-(\theta_{i}+\delta \theta +\cos^{-1}\left(\frac{1}{\sqrt{{\left(x_{i+1}-\alpha \right)}^{2}+{\left(y_{i+1}-\beta \right)}^{2}}}\left[\begin{array}{c} x_{i+1}-\alpha \\ y_{i+1}-\beta \end{array}\right]\cdot \frac{1}{\sqrt{{x_{i+1}}^{2}+{y_{i+1}}^{2}}}\left[\begin{array}{c} x_{i+1}\\ y_{i+1} \end{array}\right]\right)) \end{array} \end{array}$$
(16)$$\begin{array}{c} \begin{array}{cc} \left[\begin{array}{c} \delta x_{i}\\ \delta y_{i}\\ \delta \phi_{i} \end{array}\right] & =\left[\begin{array}{c} x_{i+1}-r_{i}\cos\theta_{i}\\ y_{i+1}-r_{i}\sin\theta_{i}\\ \phi_{i+1}-\phi_{i} \end{array}\right]\\ \; & =\left[\begin{array}{c} x_{i+1}-r_{i}\cos\theta_{i}\\ y_{i+1}-r_{i}\sin\theta_{i}\\ \frac{\pi }{2}-(\theta_{i}+\delta \theta +\cos^{-1}(\frac{1}{\sqrt{{\left(x_{i+1}-\alpha \right)}^{2}+{\left(y_{i+1}-\beta \right)}^{2}}}\left[\begin{array}{c} x_{i+1}-\alpha \\ y_{i+1}-\beta \end{array}\right]\cdot \frac{1}{\sqrt{{x_{i+1}}^{2}+{y_{i+1}}^{2}}}\left[\begin{array}{c} x_{i+1}\\ y_{i+1} \end{array}\right]))-\phi_{i} \end{array}\right] \end{array} \end{array}$$

$\phi_{i+1}$ in Figure 10b can be calculated using Equations (Equation 14) and (Equation 15). The incremental vector δ$x_{i}$, δ$y_{i}$, δ$\phi_{i}$, which is the incremental position of the ultrasound probe, can be calculated using Equations (Equation 15) and (Equation 16).
(17)$$\begin{array}{c} \begin{array}{cc} \left[\begin{array}{c} \delta r_{i}\\ \delta \theta \\ \delta \psi_{i} \end{array}\right] & ={J_{n}}^{-1}\left[\begin{array}{c} \delta x_{i}\\ \delta y_{i}\\ \delta \phi_{i} \end{array}\right]+\left[\begin{array}{c} 0\\ \delta \theta \\ 0 \end{array}\right]\\ \; & =-\frac{1}{\left(r_{i}+H\right)}\left[\begin{array}{ccc} -\left(r_{i}+H\right)\cos\theta_{i} & -\left(r_{i}+H\right)\sin\theta_{i} & -\left(r_{i}+H\right)H\sin\psi_{i}\\ 0 & 0 & 0\\ -\sin\theta_{i} & \cos\theta_{i} & \left(r_{i}+H\right)-H\sin\left(\psi_{i}+2\theta_{i}\right) \end{array}\right]\\ \; & \left[\begin{array}{c} x_{i+1}-r_{i}\cos\theta_{i}\\ \tan\left(\theta_{i}+\delta \theta \right)x_{i}-r_{i}\sin\theta_{i}\\ \frac{\pi }{2}-(\theta_{i}+\delta \theta +\cos^{-1}(\frac{1}{\sqrt{{\left(x_{i+1}-\alpha \right)}^{2}+{\left(y_{i+1}-\beta \right)}^{2}}}\left[\begin{array}{c} x_{i+1}-\alpha \\ y_{i+1}-\beta \end{array}\right]\cdot \frac{1}{\sqrt{{x_{i+1}}^{2}+{y_{i+1}}^{2}}}\left[\begin{array}{c} x_{i+1}\\ y_{i+1} \end{array}\right]))-\phi_{i} \end{array}\right]\\ \; & +\left[\begin{array}{c} 0\\ \delta \theta \\ 0 \end{array}\right] \end{array} \end{array}$$

Equation (Equation 17) represents the last equation for calculating δ$r_{i}$, δθ, and δ$\psi_{i}$ vector by using Equations (Equation 5) and (Equation 16), which represents the inverse Jacobian. Note that $J_{n}^{-1}$ is calculated by nullifying second row in Equation (Equation 5).
(18)$$\begin{array}{c} \begin{array}{cc} X & =\cos^{-1}(\frac{1}{\sqrt{{\left(x_{i+1}-\alpha \right)}^{2}+{\left(y_{i+1}-\beta \right)}^{2}}}\left[\begin{array}{c} x_{i+1}-\alpha \\ y_{i+1}-\beta \end{array}\right]\cdot \frac{1}{\sqrt{{x_{i+1}}^{2}+{y_{i+1}}^{2}}}\left[\begin{array}{c} x_{i+1}\\ y_{i+1} \end{array}\right])\\ \; & =\cos^{-1}(\frac{x_{i+1}^{2}+y_{i+1}^{2}-\alpha x_{i+1}-\beta y_{i+1}}{\sqrt{\left(\begin{array}{c} x_{i+1}^{4}+y_{i+1}^{4}+2x_{i+1}^{2}y_{i+1}^{2}-2\alpha x_{i+1}^{3}\\ -2\beta y_{i+1}^{2}-2\alpha x_{i+1}y_{i+1}^{2}-2\beta x_{i+1}^{2}y_{i+1}\\ +\left(\alpha^{2}+\beta^{2}\right)x_{i+1}^{2}+\left(\alpha^{2}+\beta^{2}\right)y_{i+1}^{2} \end{array}\right)}}) \end{array} \end{array}$$
(19)$$\left[\begin{array}{c} \delta r_{i}\\ \delta \theta \\ \delta \psi_{i} \end{array}\right]=\left[\begin{array}{c} \left(\begin{array}{c} \cos\theta_{i}x_{i+1}+\sin\theta_{i}\tan\left(\theta_{i}+\delta \theta \right)x_{i}-r_{i}\\ -\frac{\pi }{2}H\left(\theta_{i}+\delta \theta \right)\sin\psi_{i}-H\left(\theta_{i}+\delta \theta \right)\sin\psi_{i}-X\sin\psi_{i}-H\phi_{i}\sin\psi_{i} \end{array}\right)\\ \delta \theta \\ \left(\begin{array}{c} \frac{\sin\theta_{i}}{\left(r_{i}+H\right)}x_{i+1}-\frac{\cos\theta_{i}\tan\left(\theta_{i}+\delta \theta \right)}{\left(r_{i}+H\right)}x_{i}+\frac{H\pi \sin(\psi_{i}+2\theta_{i})}{2(r_{i}+H)}+\frac{H\sin\left(\psi_{i}+2\theta_{i}\right)\left(\theta_{i}+\delta \theta \right)}{\left(r_{i}+H\right)}\\ -\frac{H\sin(\psi_{i}+2\theta_{i})X}{\left(r_{i}+H\right)}+\frac{H\sin\left(\psi_{i}+2\theta_{i}\right)\phi_{i}}{\left(r_{i}+H\right)}-\frac{\pi }{2}+\theta_{i}+\delta \theta -X+\phi_{i} \end{array}\right) \end{array}\right].$$

If the inverse cosine term in Equation (Equation 17) is sorted and replaced by X in Equation (Equation 18), Equation (Equation 17) can be transformed into Equation (Equation 19). Using Equation (Equation 19), the sarcopenia detection system can move the ultrasound probe from $P_{i}$ to $P_{i+1}$.

#### 2.2.3. Control Flow of the Sarcopenia Detection System

The overall control flow of the developed sarcopenia detection system is shown in Figure 11. The green-colored arrows indicate the sequential work flow direction, and the other-colored arrows indicate the respective signal directions, above which the signal names are attached. z, ψ, θ, and r indicate the four directional position setting value for the sarcopenia detection system. Note that $\psi_{\_fb}$ indicates the ψ feedback angle from the AHRS sensor in Figure 4 and $F_{r\_fb}$ indicates the r-directional force feedback from the F/T sensor in Figure 4. $F_{r\_ref}$ is the setting force value. The Jacobian-based ultrasound probe moving and r-directional force feedback control in the grey box in Figure 11 iterates until an arc scanning is completed. The piecewise arc curve fitting by Algorithm A1 block in the dark grey box in Figure 11 iterates until every arc scanning is completed. The overall sampling rate of the system was set to 50 Hz.

## 3. Results

The proposed angular thigh scanning method for the developed sarcopenia detection system was validated by an in-vitro test with a human thigh mimicked ham-gelatin phantom. Ham is thought to be a good substitute for muscle in the human thigh, and the gelatin is widely used for ultrasound phantom development [48,49]. To construct a human thigh mimicked phantom, preliminary tests were performed for measuring the stiffness of the human thigh. Each tests presented in subsequent sections.

### 3.1. Human Thigh Stiffness Measurement

To estimate the stiffness of the phantom to that of a human thigh, stiffness of the human thigh was measured with four male (age = 28 ± 3 years, height = 176 ± 4.2 cm, weight = 74 ± 6.2 kg) and three female (age = 24 ± 3 years, height = 165 ± 2.2 cm, weight = 64 ± 4.2 kg) subjects.

Stiffness was measured using the sarcopenia detection system. The configuration of the system was set to θ = 90° and ψ = 0°, which is shown in Figure 1b. Then, the subject’s thigh was inserted into the system and was pushed with the ultrasound probe by −10 mm r directonal movement from zero force position. The overall test is shown in Figure 12. During the push, the r-directional force was measured and the stiffness k (N/mm) was determined by Equation (Equation 20).
(20)$$k=\frac{F_{r\_fb}^{f}-F_{r\_fb}^{i}}{10.0}.$$

$F_{r\_fb}^{f}$ and $F_{r\_fb}^{i}$ in Equation (Equation 20) represents the final r-directional force feedback value and the initial r-directional force feedback value, respectively. Note that Equation (Equation 20) represents the average stiffness. Table 2 summarizes the stiffness test results with an average value of 0.621 (N/mm). After 3%, 5%, and 10% gelatin phantom stiffness tests, 3% gelatin with a stiffness of 0.676 (N/mm) was selected as the gelatin mass % in gelatin and distilled water mixture for the ham-gelatin phantom.

### 3.2. In-Vitro Phantom Test

#### 3.2.1. Ham-Gelatin Phantom

The ham-gelatin phantom for the in-vitro test is depicted in Figure 13a–c. A pig saving box was used for the mold to make the phantom, as depicted in Figure 13d. It is cylindrical with width of 130 mm and length of 210 mm, which is an approximate size of the human thigh. A ham was cut 110 × 110 × 20 size to fix in the mold and investigate its muscle thickness variation during an angular scanning test. In the ham-gelatin phantom, thin metal wire is used to fix the ham in the gelatin and distilled water mixture, as depicted in Figure 13a–c.

#### 3.2.2. In-Vitro Ham-Gelatin Phantom Test Result

The phantom is inserted in the sarcopenia detection system as depicted in Figure 14a,b. Note that a wet tissue is placed on the phantom because the RGB-D camera cannot measure the depth from the transparent gelatin mass. The scanning surface is determined from the depth data, as depicted in Figure 14d, which is 105.4 mm from the origin in the Z direction.

The 1 arc curve fitting result of the phantom at z = 105.4 mm is depicted in Figure 15 and Table 3. A total 29 points are extracted from the depth points and the total RMS error and the average RMS error are 5.28 and 0.18, respectively.

With the curve fitting results, ultrasound scanning was performed using Jacobian-based ultrasound probe moving with r-directional force feedback control. Note that the r-directional feedback force is gravity-compensated because the weight contribution of the ultrasound probe to the r-directional force depends on the θ angle. The scanning started at θ = 45° and ended at θ = 127°, which is the maximum allowable θ range with ±30°ψ range. The two black lines in Figure 15 represent the θ scanning range. The $F_{r\_ref}$ in Figure 11, which is the r-directional force setting value, is set to 2.5 N based on the preliminary test results. Further, δθ in Equation (Equation 19) is set to 0.5°.

The scanning sequence are depicted in Figure 16a–d. The pictures represent canonical scanning process, which are captured from movies obtained during scanning. The r-directional force, r, and ψ values with respect to θ during scanning are shown in Figure 17a–c. Note that the calculation values (r_calculation and ψ_calculation) in Figure 17b,c, respectively are the calculated values by Equation (Equation 19). In Figure 17a, the average r-directional force is 2.51 N, which is almost the same as the r-directional force setting value of 2.5 N for the feedback control. The difference between the r_measurement and r_calculation values in Figure 17b is due to the r-directional force feedback control to maintain the force at 2.5 N. The average difference between the ψ_measurement and ψ_calculation values in Figure 17c is 0.11°.

The ultrasound image was recorded during scanning with SimpliVue software [44]. The image sequences with respect to the θ angle are depicted in Figure 18a–h. The red line in Figure 18c–h indicates the muscle boundary to measure muscle thickness. Note that in Figure 18a,b, the muscle boundary is not clear. The muscle thickness was manually measured by functionality in SimpliVue software and are summarized in Table 4. Four operator measured the thickness manually and the average value is 26.01 mm with a standard deviation of 1.00 mm, which is thought to be a feasible result for clinical trials. Note that there are slight measurement differences between operators in Table 4.

## 4. Conclusions and Discussions

### 4.1. Conclusions

In this study, a sarcopenia detection system with an RGB-D camera was developed to scan human upper thigh with a commercial ultrasound probe and to diagnose sarcopenia by muscle thickness measurement from ultrasound images. To move the ultrasound probe angularly along the convex surface of the thigh, a piecewise arc curve fitting method and a Jacobian-based ultrasound probe moving method were proposed. The proposed angular thigh scanning method with the sarcopenia detection system was verified by an in-vitro test with a ham-gelatin phantom. The ultrasound scanning results of the ham-gelatin phantom show that the radial directional force of the ultrasound probe was maintained at ±0.1 N and the muscle thickness of the ham in the phantom is maintained at 25.97 ± 1.09 mm during 82° angular scanning. These results are thought to be promising for clinical trials of the proposed system and method.

### 4.2. Clinical Relevance

The RGB-D camera is fixed outside the sarcopenia detection system to match the origin of the camera coordinates and that of the system coordinates. However, in real clinical situations, such camera settings may block the patient or clinician. In the next version, the RGB-D camera must be attached to the upper part of the sarcopenia detection system to render the entire system compact. Moreover, a fixed camera has limited scanning range compared to a movable camera, which further suggests attaching the camera to the system.

The roll angle (ψ) range is ±30°, which limits the angular scanning (θ) range, as in the ham-phantom in-vitro test. A larger roll angle range is desirable to increase the angular scanning range and be compatible with any clinical situations. The width of the thigh inserted in the lower part of the system is limited to 200 mm, which is not a broad width to accommodate any clinical situations. Hence, wider and slim design is suitable.

In this study, only an in-vitro ham-gelatin phantom test was performed to verify the proposed method. Clinical trials are scheduled to address the clinical feasibility of the proposed system and method in future plans. In clinical trials, the ultrasound image acquired by the system and that acquired by clinicians must be compared to verify compatibility in clinical situations. Moreover, the overall scanning time must be considered to strengthen the clinical applicability of the proposed system.

The developed system was for measuring the transversal muscle thicknesses and areas of a human thigh. In the author’s perspective, the system can be applied to scan a human arm, which has smaller volume than a human thigh. However, the transversal muscle area of the human arm can be more sensitive to outer pressure, a more sophisticated pressure control might be desirable.   

## Figures and Tables

**Figure 1 sensors-20-04447-f001:**
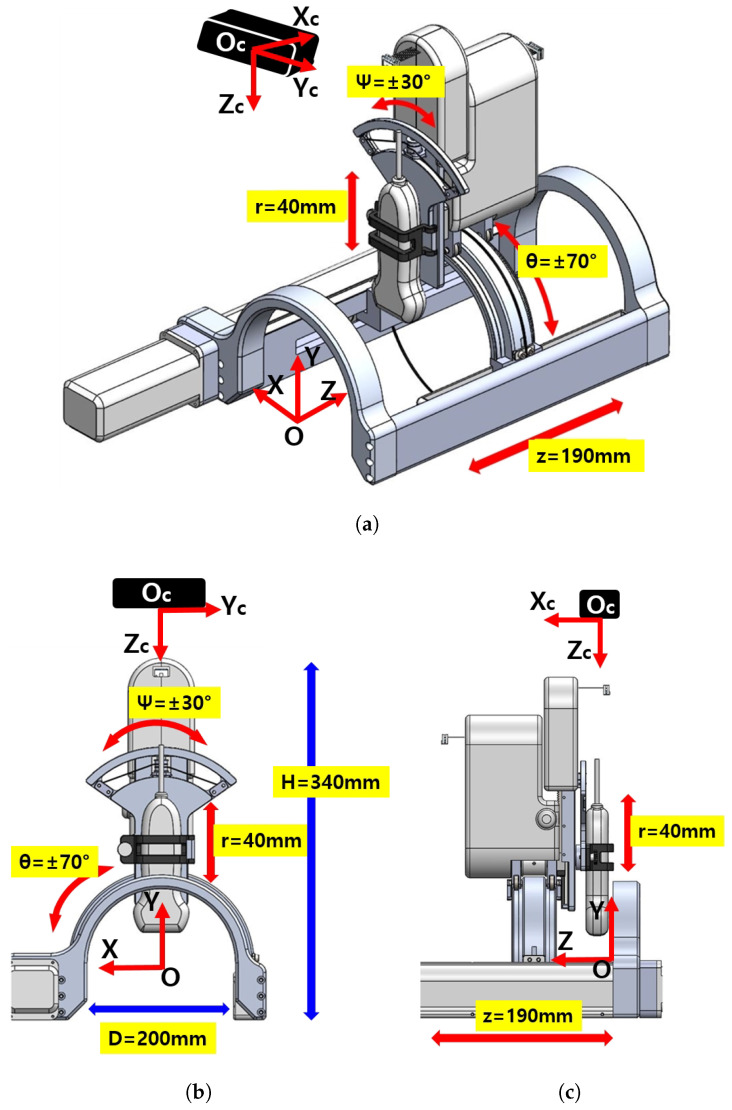
Overview of the developed 4 degrees of freedom (DOF) sarcopenia detection system. (**a**) Perspective view. (**b**) Front view. (**c**) Side view.

**Figure 2 sensors-20-04447-f002:**
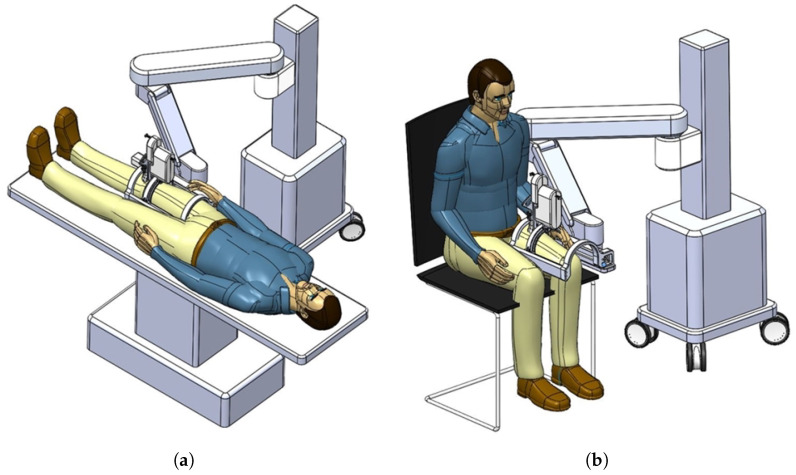
Usages of the developed sarcopenia detection system. (**a**) Lay down usage. (**b**) Sit down usage.

**Figure 3 sensors-20-04447-f003:**
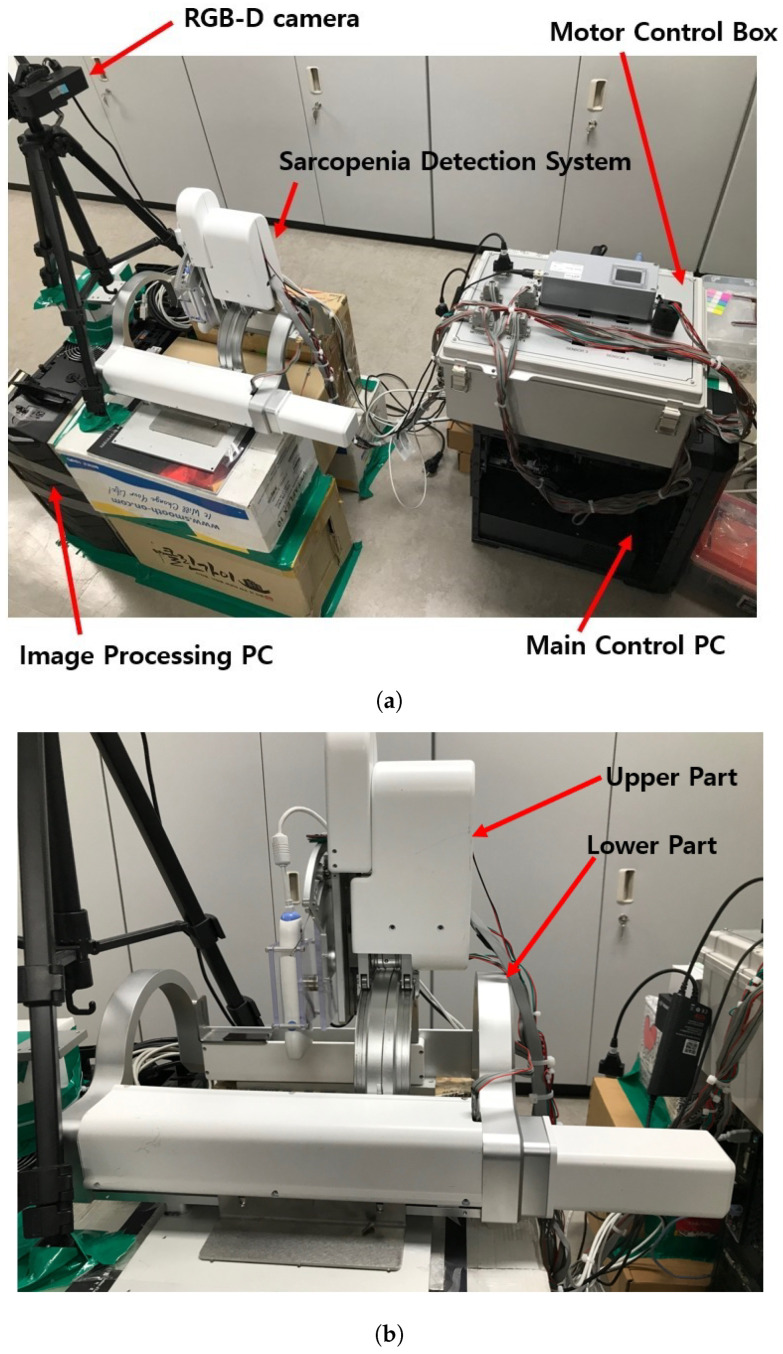
Test section and sarcopenia detection system. (**a**) Test section. (**b**) Sarcopenia detection system.

**Figure 4 sensors-20-04447-f004:**
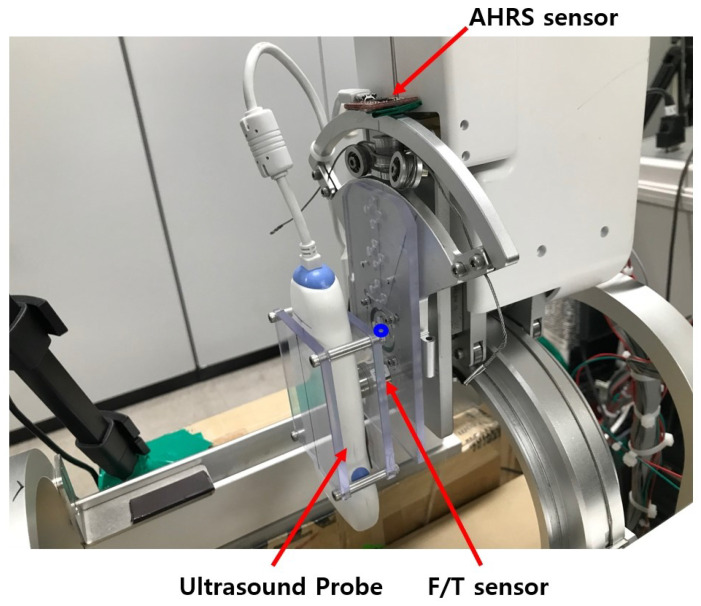
Sensors in the upper part of sarcopenia detection system.

**Figure 5 sensors-20-04447-f005:**
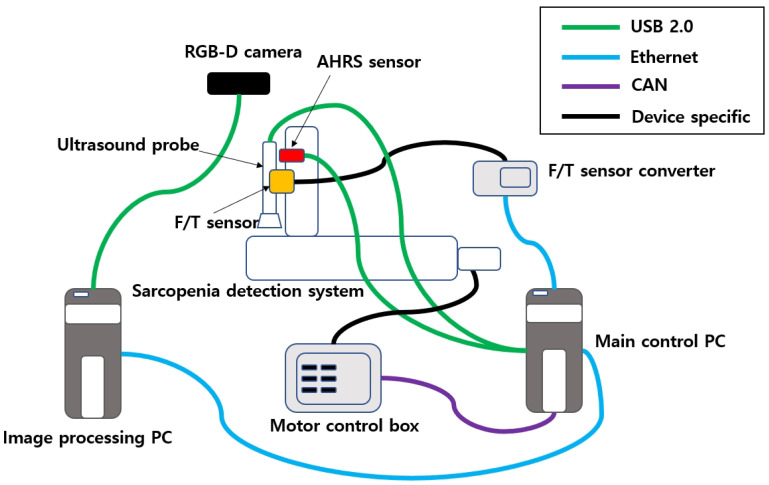
Communication schematics of the sarcopenia detection system.

**Figure 6 sensors-20-04447-f006:**
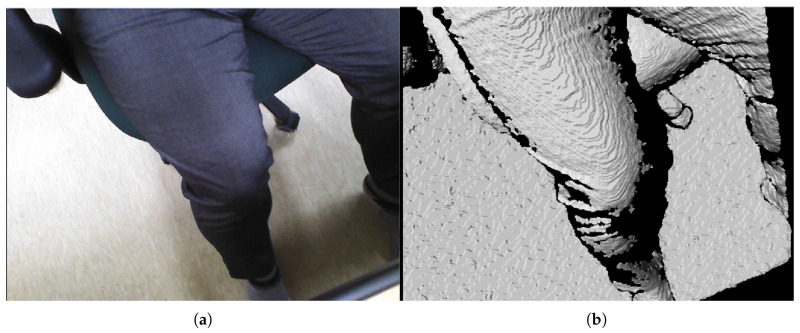
Point cloud example by RGB-D camera [41]. (**a**) Point cloud example (RGB data). (**b**) Point cloud example (D data).

**Figure 7 sensors-20-04447-f007:**
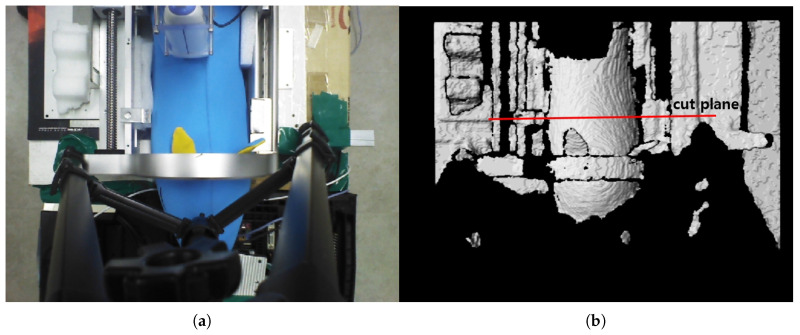
RGB and D data by the data acquisition program [41] after inserting a pig doll in the sarcopenia detection system. (**a**) RGB data. (**b**) D data.

**Figure 8 sensors-20-04447-f008:**
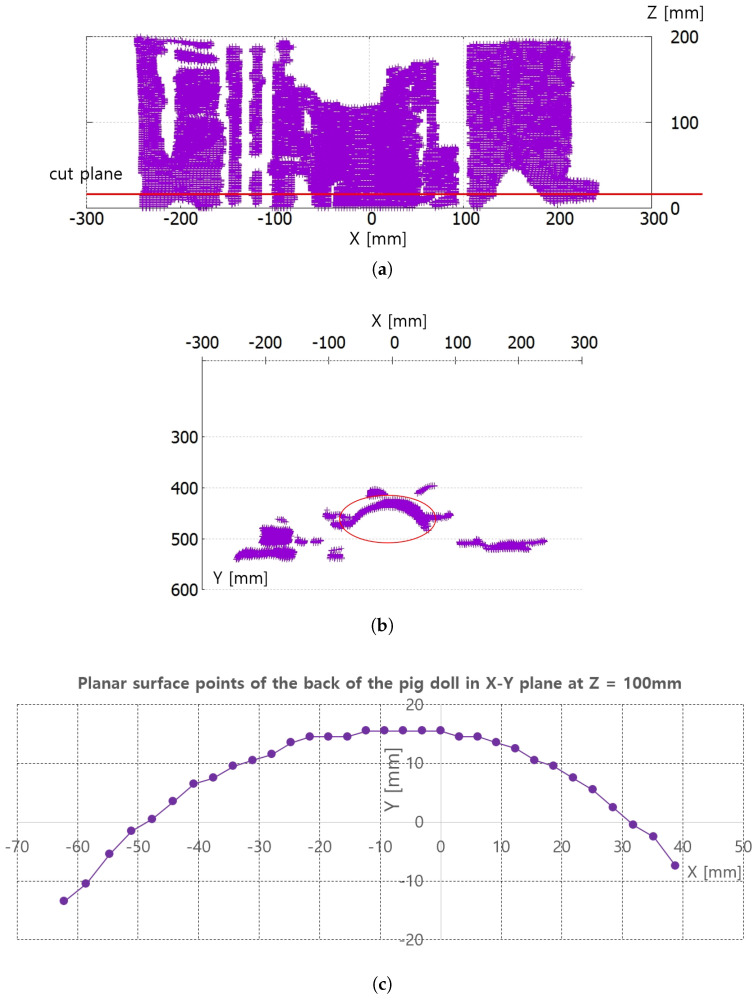
Points extraction from RGB-D data in Figure 7 for piecewise arc curve fitting. (**a**) X-Z plane view of all the D data points. (**b**) X-Y plane view of all the D data points. (**c**) Extracted points of back surface of the pig doll in X-Y plane at Z = 100 mm.

**Figure 9 sensors-20-04447-f009:**
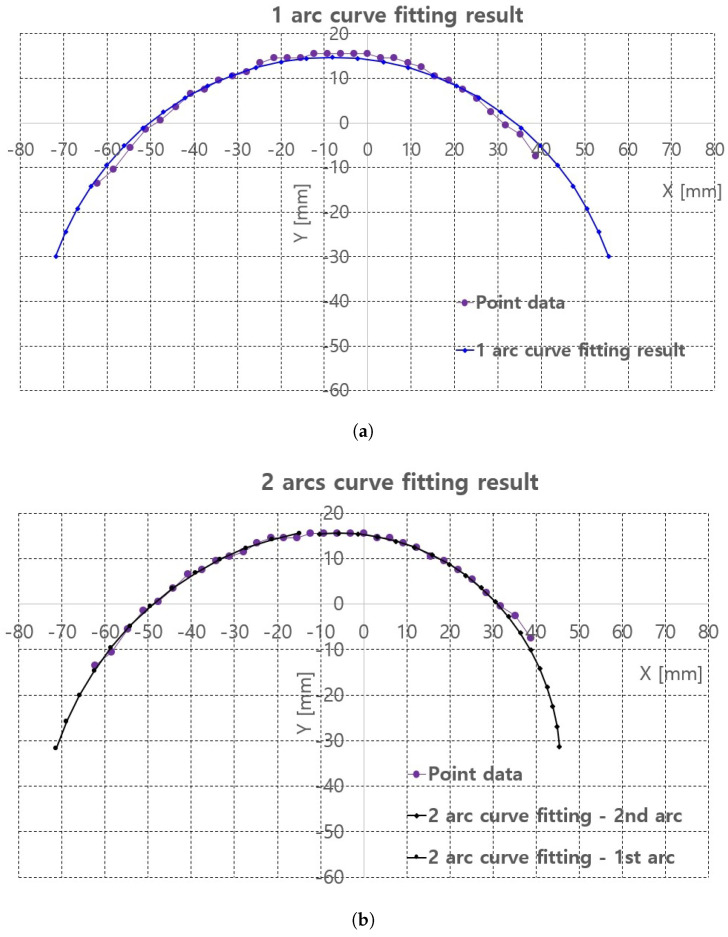
Piecewire curve fitting results of extracted points of back surface of the pig doll. (**a**) 1 arc curve fitting result. (**b**) 2 arc curve fitting result.

**Figure 10 sensors-20-04447-f010:**
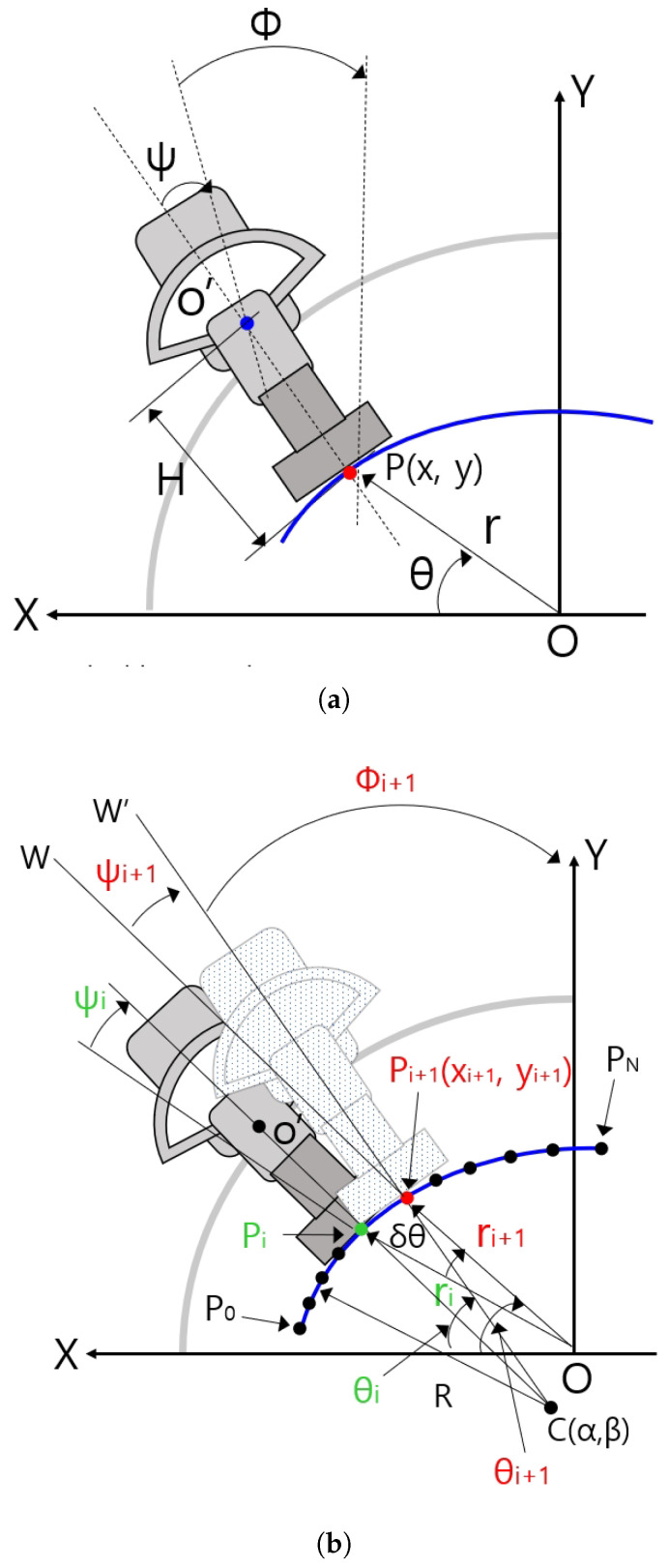
Geometric variables and relations in sarcopenia detection system. (**a**) Geometric variables for planar kinematics of the sarcopenia detection system. (**b**) Geometric variables for Jacobian based ultrasound scanning method.

**Figure 11 sensors-20-04447-f011:**
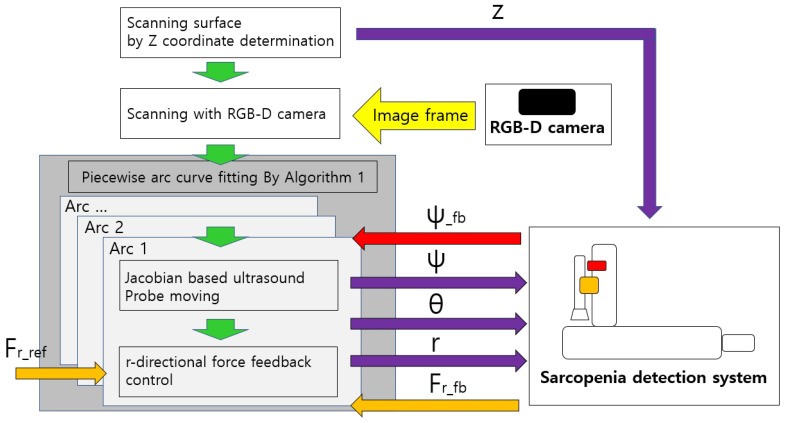
Overall control flow diagram of sarcopenia detection system—angular (planar) thigh scanning.

**Figure 12 sensors-20-04447-f012:**
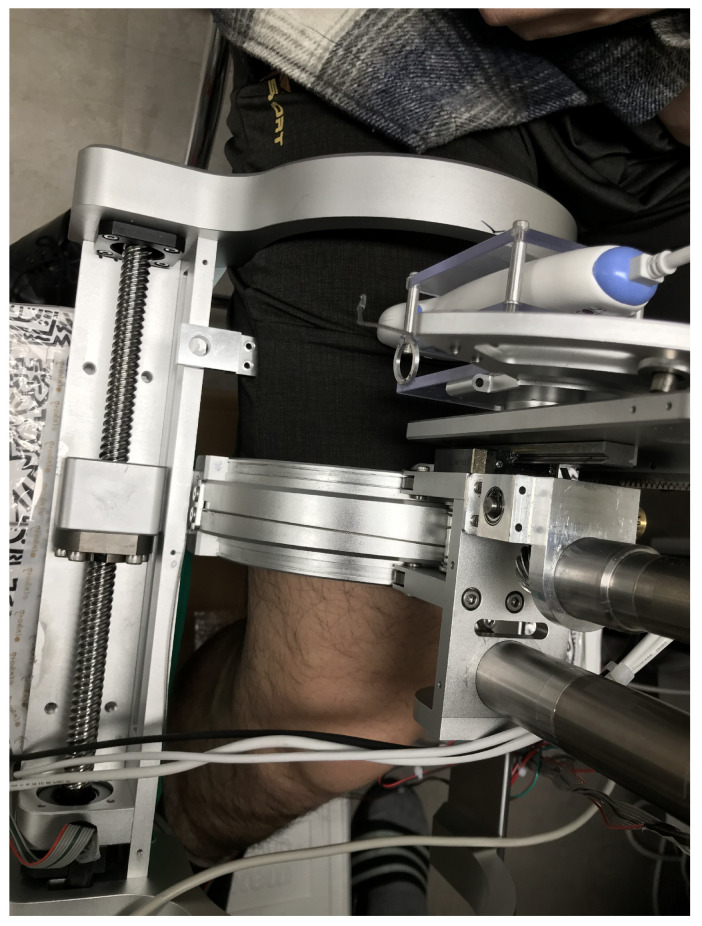
Stiffness test picture.

**Figure 13 sensors-20-04447-f013:**
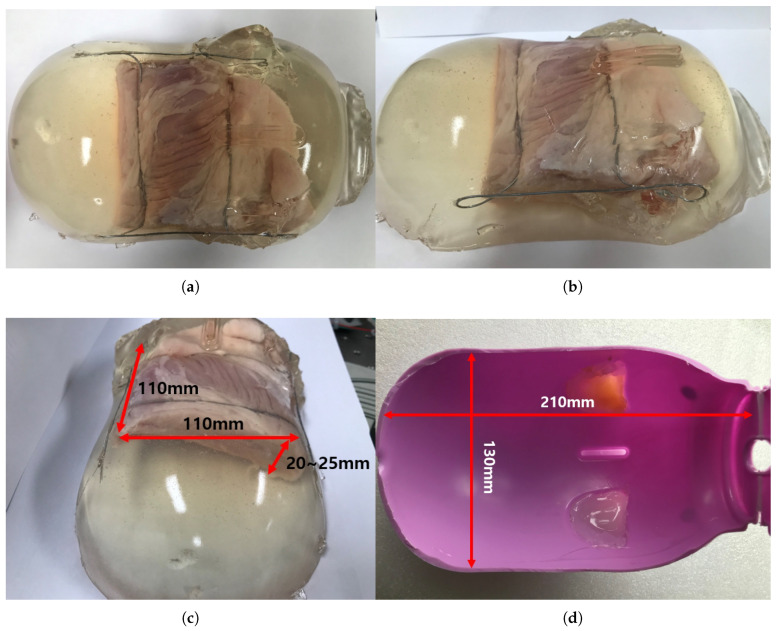
Ham-gelatin phantom for in-vitro test. (**a**) Ham-gelatin phantom (Plane view). (**b**) Ham-gelatin phantom (Side view). (**c**) Ham-gelatin phantom (Rear view). (**d**) Mold for Ham-gelatin phantom.

**Figure 14 sensors-20-04447-f014:**
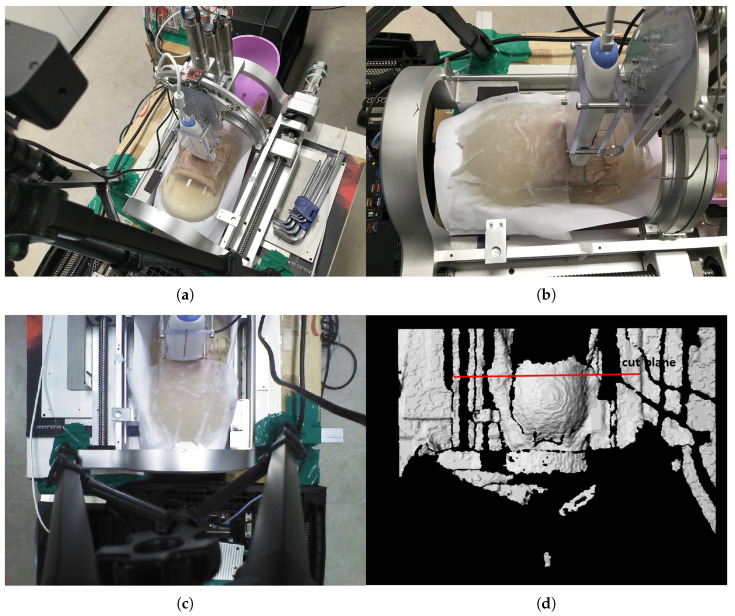
In-vitro test pictures. (**a**) Ham-gelatin phantom inserted in the sarcopinia detection system (Perspective view). (**b**) Ham-gelatin phantom inserted in sarcopinia detection system (Side view). (**c**) RGB data. (**d**) D data.

**Figure 15 sensors-20-04447-f015:**
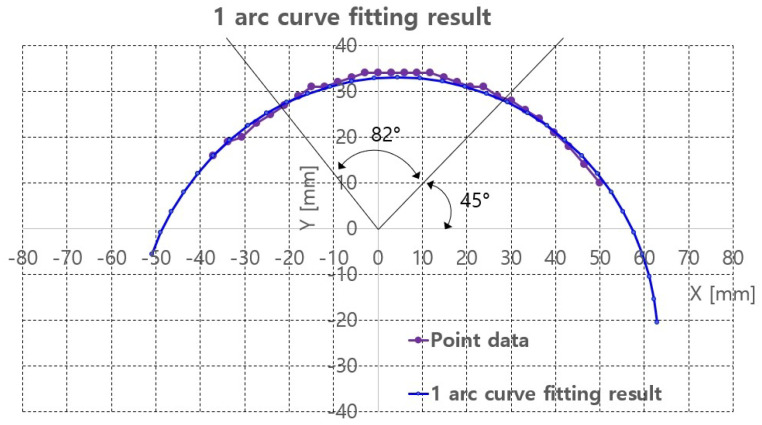
One arc curve fitting results of extracted points of the phantom at z = 105.4 mm.

**Figure 16 sensors-20-04447-f016:**
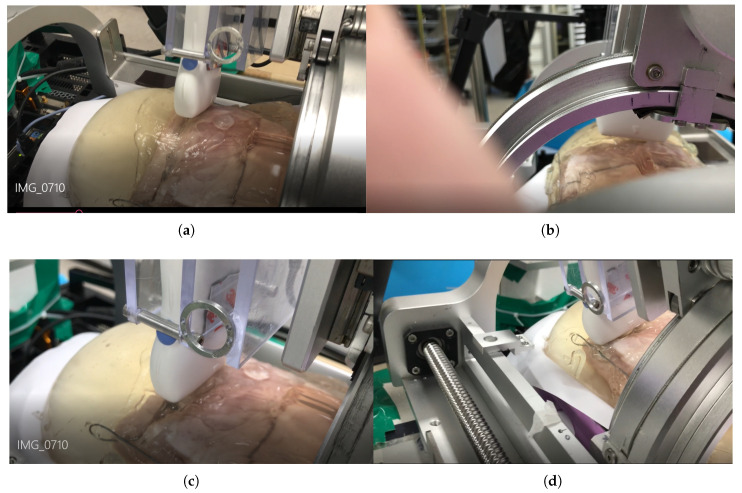
Phantom ultrasound scanning sequences. (**a**) Initial scanning position. (**b**) 1/3 scanning position. (**c**) 2/3 scanning position. (**d**) Final scanning position.

**Figure 17 sensors-20-04447-f017:**
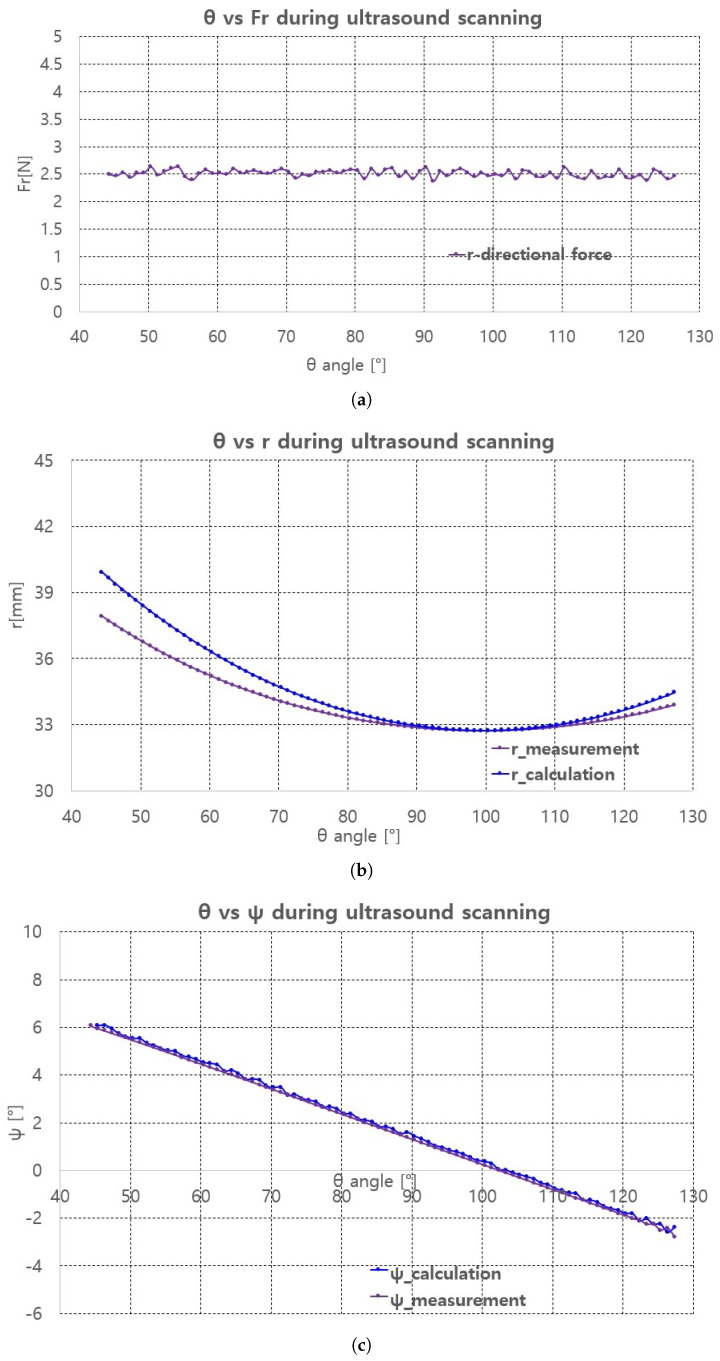
Fr, r, ψ measurement and calculation with respect to θ. (**a**) r-directional force measurement. (**b**) r measument and calculation. (**c**) ψ measument and calculation.

**Figure 18 sensors-20-04447-f018:**
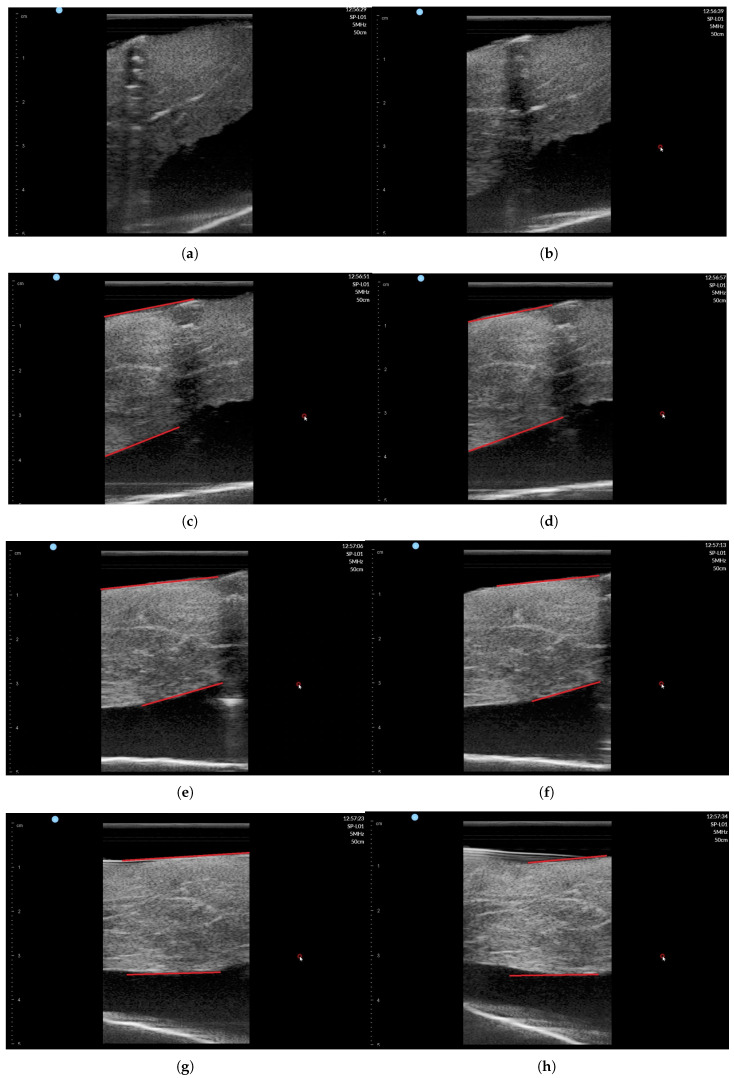
Phantom ultrasound scanning sequences with operator #1 measurement. (**a**) θ = 50°. (**b**) θ = 60°. (**c**) θ = 70°. (**d**) θ = 80°. (**e**) θ = 90°. (**f**) θ = 100°. (**g**) θ = 110°. (**h**) θ = 120°.

**Table 1 sensors-20-04447-t001:** Piecewise arc curve fitting results (mm).

		Radius of Arc	Center Point of Arc (x,y)		Total Error	Average Error
1 arc curve fitting		67.77	−8.16	−53.06	6.34	0.19
2 arc curve fitting	1st arc	73.62	−2.20	−56.89	1.22	0.04
	2nd arc	51.42	−5.79	−35.85		

**Table 2 sensors-20-04447-t002:** Stiffness test results (N/mm).

Stiffness	Male				Female			Average
	Subject 1	Subject 2	Subject 3	Subject 4	Subject 5	Subject 6	Subject 7	
	0.602	0.902	0.886	0.269	0.497	0.848	0.344	0.621

**Table 3 sensors-20-04447-t003:** Piecewise curve fitting result of the phantom (mm).

	Radius of Arc	Center Point of Arc (x,y)		Total Error	Average Error
1 arc curve fitting	58.75	4.37	−25.65	5.28	0.18

**Table 4 sensors-20-04447-t004:** Muscle thickness measurement results.

	θ Angle (°)	50	60	70	80	90	100	110	120	Average	Std. Deviation
Operator #1	muscle thickness (mm)	-	-	27.54	26.41	24.52	24.53	26.41	26.43	25.97	1.09
muscle thickness- average value (mm)	-	-	1.57	0.44	−1.44	−1.43	0.44	0.46	
Operator #2	muscle thickness (mm)	-	-	27.47	26.45	24.9	24.56	26.11	26.22	25.95	0.97
muscle thickness- average value (mm)	-	-	1.51	0.49	−1.05	−1.38	0.15	0.27	
Operator #3	muscle thickness (mm)	-	-	27.73	26.86	24.56	24.64	25.84	25.47	25.85	1.14
muscle thickness- average value (mm)	-	-	1.88	1.01	−1.28	−1.21	0.00	−0.38	
Operator #4	muscle thickness (mm)	-	-	27.73	27.16	24.9	25.66	25.84	26.22	26.25	0.94
muscle thickness- average value (mm)	-	-	1.47	0.91	−1.35	−0.59	−0.40	−0.03	
Average	muscle thickness (mm)	-	-	27.62	26.72	24.72	24.84	26.05	26.08	26.01	1.00
muscle thickness- average value (mm)	-	-	1.61	0.71	−1.28	−1.16	0.04	0.07

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
