# Peer review of "Sarcopenia Detection System Using RGB-D Camera and Ultrasound Probe: System Development and Preclinical In-Vitro Test"

_sensors, 2020, doi:10.3390/s20164447_

Round 1
Reviewer 1 Report
The authors present a system for the detection of sarcopenia using an RGB-D camera coupled with an ultrasound probe. The paper has merit and it is well written. It is also very clear how the hardware works in the proposed system. My comments as a Review are listed as follows:
1) When talking about quantitative muscle ultrasound (page 2, line 52), the following paper should be cited (10.1016/j.ultrasmedbio.2016.08.032). In these works, the authors proposed an automated algorithm for the analysis of muscles in both longitudinal and transversal plane (automatic muscle thickness measurement and cross-sectional area evaluation).
2) Has muscle thickness (figure 18) been measured manually? Since the presence of inter- and intra-operator variability, the authors should repeat the measures reported in Table 4 for at least two different manual operators.
3) Future works: in your opinion, can this system be applied to other superficial muscles of the lower/upper limb?
Minor comments:
- page 2, line 38: typo “Ultrasouud”
- page 14, line 263 and page 16, line 284: duplicated subsection title
- page 14, line 269: missing subsection number
Author Response
The authors present a system for the detection of sarcopenia using an RGB-D camera coupled with an ultrasound probe. The paper has merit and it is well written. It is also very clear how the hardware works in the proposed system. My comments as a Review are listed as follows:
At first, thank you for kind and thorough review of the submitted manuscript. Your reviews and comments are valuable and made the manuscript more complete and understandable. The answers of the comments are written in blue-colored sentences and the changes made in the manuscript were also written in blue-colored texts.
1) When talking about quantitative muscle ultrasound (page 2, line 52), the following paper should be cited (10.1016/j.ultrasmedbio.2016.08.032). In these works, the authors proposed an automated algorithm for the analysis of muscles in both longitudinal and transversal plane (automatic muscle thickness measurement and cross-sectional area evaluation).
- Thank you for the additional reference. We added 10.1016/j.ultrasmedbio.2016.08.032 reference in Page 2, line 52~55 as reference [26].
2) Has muscle thickness (figure 18) been measured manually? Since the presence of inter- and intra-operator variability, the authors should repeat the measures reported in Table 4 for at least two different manual operators.
- The muscle thickness was manually measured with commercial software. As the reviewer suggested, for the inter and intra operator variability, we added other 3 operators’ measurement results in Table 4. Please, see Table 4 in Page 28 and sentences of Page 24, line 38 ~ Page 25, line 392.
3) Future works: in your opinion, can this system be applied to other superficial muscles of the lower/upper limb?
- As the reviewer commented, we added additional comments on other superficial muscles of the upper limb in Page 28, line 423~426.
Minor comments:
- page 2, line 38: typo “Ultrasouud”
- Thank you for your careful mistype comments. We changed “Ultrasouud” to “Ultrasound”
- page 14, line 263 and page 16, line 284: duplicated subsection title
- Thank you for your comments. As you suggested, we changed the subsection title to Ultrasound Probe Moving Method and the subsubsection title to Jacobian Based Ultrasound Prove Moving. Please, see Page 14 and 17 for the changes.
- page 14, line 269: missing subsection number
Thank you for your kind comment. As you commented, we added subsection numbers (2.2.2.1. and 2.2.2.2) in Page 14 and Page 17.
Reviewer 2 Report
In this manuscript, the authors investigated a RGB camera for sarcopenia detection system. The study is well written, has interesting results, and should be of great interest to the readers of Sensors. This research was well-organized and showed positive results. I recommend accepting this manuscript after minor revision because of the following issues.
1.) Title:
The title should say something to impress readers. Please perform a new title, which gives clinical relevance of the paper.
2.) Abstract:
Well written.
3.) Introduction
This section shows clear structure and provide a complete context to the readers.
4.) Materials and methods
Very detailed and correct.
5.) Results
The results section is very appropriate according to the developed methods and the journal´s scope.
6.) Discussion.
This section is useful and improves the understanding of the results section comparing with novel and adequate studies. Please, add a subsection called "clinical relevance" or "clinical applications"
7.) Conclusion.
Please, authors should develop a conclusion section "more conclusive". The present form is too long.
Author Response
In this manuscript, the authors investigated a RGB camera for sarcopenia detection system. The study is well written, has interesting results, and should be of great interest to the readers of Sensors. This research was well-organized and showed positive results. I recommend accepting this manuscript after minor revision because of the following issues.
- At first, thank you for your kind and thorough review of the submitted manuscript. Your reviews and comments are valuable and made the manuscript more complete and understandable. The answers of the comments were written in blue-colored sentences and the changes made in the manuscript were also written in blue-colored texts.
1.) Title:
The title should say something to impress readers. Please perform a new title, which gives clinical relevance of the paper.
- Thank you for the kind comments. As you suggested, we changed the title of the manuscript to “Sarcopenia Detection System Using RGB-D Camera and Ultrasound Probe: System Development and Preclinical In-vitro Test”. We hope this would help the readers understand our works.
2.) Abstract:
Well written.
- Thank you for your kind comment.
3.) Introduction
This section shows clear structure and provide a complete context to the readers.
- Thank you for your kind comment.
4.) Materials and methods
Very detailed and correct.
- Thank you for your kind comment.
5.) Results
The results section is very appropriate according to the developed methods and the journal´s scope.
- Thank you for your kind comment.
) Discussion.
This section is useful and improves the understanding of the results section comparing with novel and adequate studies. Please, add a subsection called "clinical relevance" or "clinical applications" - Thank you for your careful comments. We divided “Results and Discussions” section to “Results” subsection and “Clinical Relevance” subsection. Please, see Page 25 ~ 28.
7.) Conclusion.
Please, authors should develop a conclusion section "more conclusive". The present form is too long.
- Thank you for your kind comment. We made “Results” subsection more conclusive and concise. Please, see Page 25, line 394~404.
Reviewer 3 Report
This paper is well-written and timely. The authors try to address a public health problem. The robot design, modeling, and experimental validation are sufficient. This paper can be accepted as it is.
Author Response
This paper is well-written and timely. The authors try to address a public health problem. The robot design, modeling, and experimental validation are sufficient. This paper can be accepted as it is.
- At first, thank you for your kind and thorough review of the submitted manuscript. Your reviews and comments are valuable and made the manuscript more complete and understandable. The answers of the comments were written in blue-colored sentences and the changes made in the manuscript were also written in blue-colored texts.
- Thank you for your kind comment.